# ON THE LATENT SPACE OF FLOW-BASED MODELS

## ABSTRACT

Flow-based generative models typically define a latent space with dimensionality identical to the observational space. In many problems, however, the data does not populate the full ambient data-space that they natively reside in, but rather inhabit a lower-dimensional manifold. In such scenarios, flow-based models are unable to represent data structures exactly as their density will always have support off the data manifold, potentially resulting in degradation of model performance. In addition, the requirement for equal latent and data space dimensionality can unnecessarily increase model complexity for contemporary flow models. Towards addressing these problems, we propose to learn a manifold prior that affords benefits to both the tasks of sample generation and representation quality. An auxiliary product of our approach is that we are able to identify the intrinsic dimension of the data distribution.

## 1 INTRODUCTION

Normalizing flows (Rezende and Mohamed, 2015; Kobyzev et al., 2020) have shown considerable potential for the tasks of modelling and inferring expressive distributions through the learning of well-specified probabilistic models. Contemporary flow-based approaches define a latent space with dimensionality identical to the data space, typically by parameterizing a complex model $p_X(x|\theta)$ using an invertible neural network $f_\theta$. Samples drawn from an initial, simple distribution $p_Z(z)$ (*e.g.* Gaussian) can be mapped to a complex distribution as $x = f_\theta(z)$. The process results in a tractable density that inhabits the full data space. However, contemporary flow models may make for an inappropriate choice to represent data that resides in a lower-dimensional manifold and thus does not populate the full ambient space. In such cases, the estimated model will necessarily have mass lying off the data manifold, which may result in under-fitting and poor generation qualities. Furthermore, principal objectives such as Maximum Likelihood Estimation (MLE) and Kullback-Leibler (KL) divergence minimization are ill-defined, bringing additional challenges for model training.

In this work, we propose a principled strategy to model a data distribution that lies on a continuous manifold and we additionally identify the intrinsic dimension of the data manifold. Specifically, by using the connection between MLE and KL divergence minimization in $Z$ space, we can address the important problem of ill-defined KL divergence under typical flow based assumptions.

Flow models are based on the idea of "change of variable". Assume a random variable $Z$ with distribution $\mathbb{P}_Z$ and probability density $p_Z(z)$. We can transform $Z$ to get a random variable $X$: $X = f(Z)$, where $f : \mathbb{R}^D \to \mathbb{R}^D$ is an invertible function with inverse $f^{-1} = g$. Suppose $X$ has distribution $\mathbb{P}_X$ and density function $p_X(x)$, then $\log p_X(x)$ will have the following form

$$\log p_X(x) = \log p_Z(g(x)) + \log \left| \det \left( \frac{\partial g}{\partial x} \right) \right|, \tag{1}$$

where $\log \left| \det \left( \frac{\partial g}{\partial x} \right) \right|$ is the log determinant of the Jacobian matrix. We call $f$ (or $g$) a *volume-preserving* function if the log determinant is equal to 0.

Training of flow models typically makes use of MLE. We denote $X_d$ as the random variable of the data with distribution $\mathbb{P}_d$ and density $p_d(x)$. In addition to the well-known connection between MLE and minimization of the KL divergence $\mathrm{KL}(p_d(x)||p_X(x))$ in $X$ space (see Appendix A for detail), MLE is also (approximately) equivalent to minimizing the KL divergence in $Z$ space, this is due to the KL divergence is invariant under invertible transformation (Yeung, 2008; Papamakarios et al.,

2019). Specifically, we define $Z_{\mathbb{Q}} : Z_{\mathbb{Q}} = g(X_d)$ with distribution $\mathbb{Q}_Z$ and density function $q(z)$, the KL divergence in $Z$ space can be written as

$$\mathrm{KL}(q(z)||p(z)) = \int q(z) \log q(z)dz - \int q(z) \log p(z)dz \tag{2}$$

$$= -\int p_d(x) \left( \log p_Z\left(g(x)\right) + \log \left|\det\left(\frac{\partial g}{\partial x}\right)\right| \right) dx + const., \tag{3}$$

The full derivation can be found in Appendix A. Since we can only access samples $x_1, x_2, \ldots, x_N$ from $p_d(x)$, we approximate the integral by Monte Carlo sampling

$$\mathrm{KL}(q(z)||p(z)) \approx -\frac{1}{N} \sum_{n=1}^{N} \log p_X(x_n) + const.. \tag{4}$$

We thus highlight the connection between MLE and KL divergence minimization, in $Z$ space, for flow based models. The prior distribution $p(z)$ is usually chosen to be a $D$-dimensional Gaussian distribution. However, if the data distribution $\mathbb{P}_d$ is singular, for example a measure on a low dimensional manifold, the induced latent distribution $\mathbb{Q}_Z$ is also singular. In this case, the KL divergence in equation 2 is typically not well-defined under the considered flow based model assumptions. This issue brings both theoretical and practical challenges that we will discuss in the following section.

## 2 FLOW MODELS WITH MANIFOLD DATA

We assume a data sample $\mathbf{x} \sim \mathbb{P}_d$ to be a $D$ dimensional vector $\mathbf{x} \in \mathbb{R}^D$ and define the ambient dimensionality of $\mathbb{P}_d$, denoted by $\mathtt{Amdim}(\mathbb{P}_d)$, to be $D$. However for many datasets of interest, *e.g.* natural images, the data distribution $\mathbb{P}_d$ is commonly believed to be supported on a lower dimensional manifold (Beymer and Poggio, 1996). We assume the dimensionality of the manifold to be $K$ where $K < D$, and define the intrinsic dimension of $\mathbb{P}_d$, denoted by $\mathtt{Indim}(\mathbb{P}_d)$, to be the dimension of this manifold. Figure 1a provides an example of this setting where $\mathbb{P}_d$ is a 1D distribution in 2D space. Specifically, each data sample $\mathbf{x} \sim \mathbb{P}_d$ is a 2D vector $\mathbf{x} = \{x_1, x_2\}$ where $x_1 \sim \mathcal{N}(0, 1)$ and $x_2 = \sin(2x_1)$. Therefore, this example results in $\mathtt{Amdim}(\mathbb{P}_d) = 2$ and $\mathtt{Indim}(\mathbb{P}_d) = 1$.

In flow-based models, function $f$ is constructed such that it is both bijective and differentiable. When the prior $\mathbb{P}_Z$ is a distribution whose support is $\mathbb{R}^D$ (*e.g.* Multivariate Gaussian distribution), the marginal distribution $\mathbb{P}_X$ will also have support $\mathbb{R}^D$ and $\mathtt{Amdim}(\mathbb{P}_X) = \mathtt{Indim}(\mathbb{P}_X) = D$. When the support of the data distribution lies on a $K$-dimensional manifold and $K < D$, $\mathbb{P}_d$ and $\mathbb{P}_X$ are constrained to have different support. That is, the intrinsic dimensions of $\mathbb{P}_X$ and $\mathbb{P}_d$ are always different; $\mathtt{Indim}(\mathbb{P}_X) \neq \mathtt{Indim}(\mathbb{P}_d)$. In this case it is impossible to learn a model distribution $\mathbb{P}_X$ identical to the data distribution $\mathbb{P}_d$. Nevertheless, flow-based models have shown strong empirical success in real-world problem domains such as the ability to generate high quality and realistic images (Kingma and Dhariwal, 2018). Towards investigating the cause and explaining this disparity between theory and practice, we employ a toy example to provide intuition for the effects and consequences resulting from model and data distributions that possess differing intrinsic dimension.

Consider the toy dataset introduced previously; a 1D distribution lying in a 2D space (Figure 1a). The prior density $p(z)$ is a standard 2D Gaussian $p(z) = \mathcal{N}(0, I_Z)$ and the function $f$ is a non-volume preserving flow with two coupling layers (see Appendix C.1). In Figure 1b we plot samples from the flow model; the sample $\mathbf{x}$ is generated by first sampling a 2D datapoint $\mathbf{z} \sim \mathcal{N}(0, I_Z)$ and then letting $\mathbf{x} = f(\mathbf{z})$. Figure 1c shows samples from the prior distributions $\mathbb{P}_Z$ and $\mathbb{Q}_Z$. $\mathbb{Q}_Z$ is defined as the transformation of $\mathbb{P}_d$ using the bijective function $g$, such that $\mathbb{Q}_Z$ is constrained to support a 1D manifold in 2D space, and $\mathtt{Indim}(\mathbb{Q}_Z) = \mathtt{Indim}(\mathbb{P}_d) = 1$. Training of $\mathbb{Q}_Z$ to match $\mathbb{P}_Z$ (which has intrinsic dimension 2), can be seen in Figure 1c to result in curling up of the manifold in the latent space, contorting it towards satisfying a distribution that has intrinsic dimension 2. This ill-behaved phenomenon causes several potential problems for contemporary flow models:

1. *Poor sample quality.* Figure 1b shows examples where incorrect assumptions in turn result in the model generating bad samples.

2. *Low quality data representations.* The discussed characteristic that results in "curling up" of the latent space may cause degradations of the representation quality.

3. *Inefficient use of network capacity.* Neural network capacity is spent on contorting the distribution $\mathbb{Q}_Z$ to satisfy imposed dimensionality constraints.

A natural solution to the problem of intrinsic dimension mismatch is to select a prior distribution $\mathbb{P}_Z$ with the same dimensionality as the intrinsic dimension of the data distribution such that: $\texttt{Indim}(\mathbb{P}_Z) = \texttt{Indim}(\mathbb{P}_d)$. However, since we do not know $\texttt{Indim}(\mathbb{P}_d)$ explicitly, one option involves to instead learn it from the data distribution. In the following section, we will introduce a parameterization approach that enables us to learn $\texttt{Indim}(\mathbb{P}_d)$.

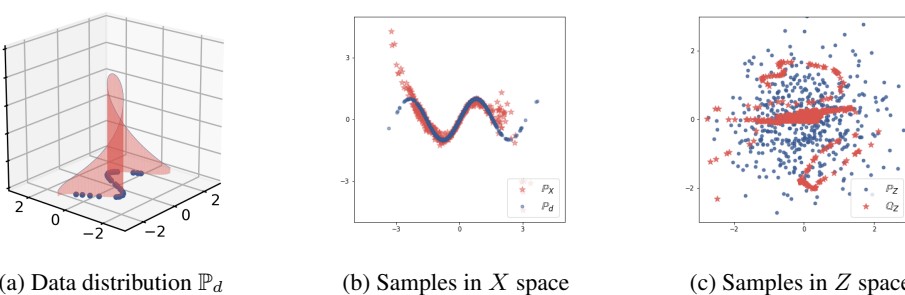

(a) Data distribution $\mathbb{P}_d$        (b) Samples in $X$ space        (c) Samples in $Z$ space

Figure 1: Samples and latent visualization from a flow based model with a fixed Gaussian prior when the intrinsic dimension is strictly lower than the true dimensionality of the data space.

## 3 LEARNING A MANIFOLD PRIOR

Consider a data vector $\mathbf{x} \in \mathbb{R}^D$, then a flow based model prior $\mathbb{P}_Z$ is usually given by a $D$-dimensional Gaussian distribution or alternative simple distribution that is also absolutely continuous (a.c.) in $\mathbb{R}^D$. Therefore, the intrinsic dimension $\texttt{Indim}(\mathbb{P}_Z) = D$. To allow a prior to have intrinsic dimension strictly less than $D$, we let $\mathbb{P}_Z$ have the '*generalized density*'[1] $p(\mathbf{z})$ with the form

$$p(\mathbf{z}) = \mathcal{N}(0, AA^T), \quad (5)$$

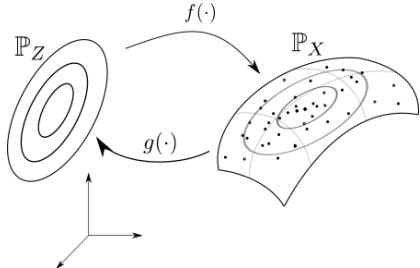

Figure 2: The data (black dots) lies on a 2D manifold in 3D space. To have a model $\mathbb{P}_X$ with $\texttt{Indim}(\mathbb{P}_X) = 2$, we learn a prior $\mathbb{P}_Z$ that is a 2D Gaussian in 3D space and an invertible function $f$ which maps from $\mathbb{P}_Z$ to $\mathbb{P}_X$.

where $\mathbf{z} \in \mathbb{R}^D$ and $A$ is a $D \times D$ lower triangular matrix with $\frac{D(D+1)}{2}$ parameters, such that $AA^T$ is constrained to be a positive semi-definite matrix. When $AA^T$ has full rank $D$, then $\mathbb{P}_Z$ is a (non-degenerate) multivariate Gaussian on $\mathbb{R}^D$. When $\texttt{Rank}(AA^T) = K$ and $K < D$, then $\mathbb{P}_Z$ will degenerate to a Gaussian supported on a $K$-dimensional manifold, such that the intrinsic dimension $\texttt{Indim}(\mathbb{P}_Z) = K$. Figure 2 illustrates a sketch of this scenario. In practice, we initialize $A$ to be an identity matrix, so $AA^T$ will also be an identity matrix and $\mathbb{P}_Z$ is initialized as a standard Gaussian.

When $\texttt{Rank}(AA^T) < D$, the degenerate covariance $AA^T$ is no longer invertible and we are unable to evaluate the density value of $p(\mathbf{z})$ for a given random vector $\mathbf{z}$. Furthermore, when the data distribution $\mathbb{P}_d$ is supported on a $K$-dimensional manifold, $\mathbb{Q}_Z$ will also be supported on a $K$-dimensional manifold and no longer has a valid density function. Using equation 2 to train the flow model then becomes impossible as the KL divergence between $\mathbb{P}_Z$ and $\mathbb{Q}_Z$ is not well defined[2]. Recent work by Zhang et al. (2020) proposed a new family of divergence to address this problem. In the following section we briefly review the key concepts pertaining to this family of divergences.

---

[1]We use the generalized density to include the case that $AA^T$ is not full rank.
[2]The KL divergence $\text{KL}(\mathbb{Q}||\mathbb{P})$ is well defined when $\mathbb{Q}$ and $\mathbb{P}$ have valid densities and their densities have the same support (Ali and Silvey, 1966).

## 4 FIX THE ILL-DEFINED KL DIVERGENCE

random Let $Z_\mathbb{Q}$ and $Z_\mathbb{P}$ be two random variables with distribution $\mathbb{Q}_Z$ and $\mathbb{P}_Z$. The KL divergence between $\mathbb{Q}_Z$ and $\mathbb{P}_Z$ is not defined if $\mathbb{Q}_Z$ or $\mathbb{P}_Z$ does not have valid density function. Let $K$ be an a.c. random variable that is independent of $Z_\mathbb{Q}$ and $Z_\mathbb{P}$ and has density $p_K$, We define $Z_{\tilde{\mathbb{P}}} = Z_\mathbb{P} + K$; $Z_{\tilde{\mathbb{Q}}} = Z_\mathbb{Q} + K$ with distributions $\tilde{\mathbb{P}}_Z$ and $\tilde{\mathbb{Q}}_Z$ respectively. Then $\tilde{\mathbb{P}}_Z$ and $\tilde{\mathbb{Q}}_Z$ are a.c. with density functions

$$q(\tilde{\mathbf{z}}) = \int_\mathbf{z} p_K(\tilde{\mathbf{z}} - \mathbf{z})d\mathbb{Q}_Z \quad p(\tilde{\mathbf{z}}) = \int_\mathbf{z} p_K(\tilde{\mathbf{z}} - \mathbf{z})d\mathbb{P}_Z. \tag{6}$$

We can thus define the *spread KL divergence* between $\mathbb{Q}_Z$ and $\mathbb{P}_Z$ as the KL divergence between $\tilde{\mathbb{Q}}_Z$ and $\tilde{\mathbb{P}}_Z$ as:

$$\widetilde{\mathrm{KL}}(\mathbb{Q}_Z||\mathbb{P}_Z) \equiv \mathrm{KL}(\tilde{\mathbb{Q}}_Z||\tilde{\mathbb{P}}_Z) \equiv \mathrm{KL}\left(q(\tilde{\mathbf{z}})||p(\tilde{\mathbf{z}})\right). \tag{7}$$

In this work we let $K$ be a Gaussian with diagonal covariance $\sigma_Z^2 I$ to satisfy the sufficient conditions such that $\widetilde{\mathrm{KL}}$ is a valid divergence (see Zhang et al. (2020) for details) and has the properties:

$$\widetilde{\mathrm{KL}}(\mathbb{Q}_Z||\mathbb{P}_Z) \geq 0, \quad \widetilde{\mathrm{KL}}(\mathbb{Q}_Z||\mathbb{P}_Z) = 0 \Leftrightarrow \mathbb{Q}_Z = \mathbb{P}_Z. \tag{8}$$

Since $\mathbb{Q}_Z$ and $\mathbb{P}_Z$ are transformed from $\mathbb{P}_d$ and $\mathbb{P}_X$ using an invertible function $g$, we have

$$\mathbb{Q}_Z = \mathbb{P}_Z \Leftrightarrow \mathbb{P}_d = \mathbb{P}_X. \tag{9}$$

Therefore, the spread KL divergence can be used to train flow based models with a manifold prior in order to fit a dataset that lies on a lower-dimensional manifold.

## 5 IDENTIFIABILITY OF INTRINSIC DIMENSION

A byproduct of our model is that the intrinsic dimension of the data manifold can be identified. Section 4 shows that when $\widetilde{\mathrm{KL}}(\mathbb{Q}_Z||\mathbb{P}_Z) = 0 \Leftrightarrow \mathbb{P}_X = \mathbb{P}_d$, the supports of $\mathbb{P}_X$ and $\mathbb{P}_d$ will also have the same intrinsic dimension: $\mathtt{Indim}(\mathbb{P}_X) = \mathtt{Indim}(\mathbb{P}_d)$. The flow function $g$, and its inverse $g = f^{-1}$, are bijective and continuous so $f$ is a diffeomorphism (Kobyzev et al., 2020). Due to the *invariance of dimension* property of diffeomorphisms (Lee, 2013, Theorem 2.17), the manifold that supports $\mathbb{P}_Z$ will have the same dimension as the manifold that supports $\mathbb{P}_X$. Therefore, we have

$$\mathtt{Indim}(\mathbb{P}_Z) = \mathtt{Indim}(\mathbb{P}_X) = \mathtt{Indim}(\mathbb{P}_d). \tag{10}$$

Since the intrinsic dimension of $\mathbb{P}_Z$ is equal to the rank of the matrix $AA^T$, we can calculate $\mathtt{Rank}(AA^T)$ by counting the number of non-zero eigenvalues of the matrix $AA^T$. This allows for identification of the intrinsic dimension of the data distribution as

$$\mathtt{Indim}(\mathbb{P}_d) = \mathtt{Rank}(AA^T). \tag{11}$$

We have shown that we can identify the intrinsic dimension of $\mathbb{P}_d$ using the spread KL divergence. In the next section, we will discuss how to estimate the spread KL divergence in practice.

## 6 ESTIMATION OF THE SPREAD KL DIVERGENCE

Our goal is to minimize the spread KL divergence between $\mathbb{Q}_Z$ and $\mathbb{P}_Z$. Using our definition of spread divergence (equation 7), this is equivalent to minimizing

$$\mathrm{KL}\left(q(\tilde{\mathbf{z}})||p(\tilde{\mathbf{z}})\right) = \underbrace{\int q(\tilde{\mathbf{z}}) \log q(\tilde{\mathbf{z}})d\tilde{\mathbf{z}}}_{\textbf{Term 1}} - \underbrace{\int q(\tilde{\mathbf{z}}) \log p(\tilde{\mathbf{z}})d\tilde{\mathbf{z}}}_{\textbf{Term 2}}. \tag{12}$$

where $q(\tilde{\mathbf{z}})$ and $p(\tilde{\mathbf{z}})$ are defined in equation 6. We separate the objective into two terms and now discuss the estimation for each of them.

**Term 1:** We use $\mathrm{H}(\cdot)$ to denote the differential entropy. Term 1 is the negative entropy $-\mathrm{H}(Z_{\tilde{\mathbb{Q}}})$. For a volume preserving $g$ and $X_d$ is a.c., the entropy $\mathrm{H}(Z_\mathbb{Q}) = \mathrm{H}(X_d)$ and is independent of the

model parameters and can be ignored during training. However, the entropy $H(Z_{\tilde{\mathbb{Q}}}) = H(Z_{\mathbb{Q}} + K)$ will still depend on $g$, see Appendix B.1 for an example. We claim that when the variance of $K$ is small, the dependency between the $H(Z_{\tilde{\mathbb{Q}}})$ and volume preserving function $g$ is weak, thus we can approximate equation 12 by leaving out term 1 and will not affect the training.

To build intuitions, we first assume $X_d$ is a.c., so $Z_{\mathbb{Q}} = g(X_d)$ is also a.c.. Using standard entropic properties (Kontoyiannis and Madiman, 2014), we can pose the following relationship

$$H(Z_{\mathbb{Q}}) \leq H(Z_{\mathbb{Q}} + K) = H(Z_{\mathbb{Q}}) + I(Z_{\mathbb{Q}} + K, K), \tag{13}$$

where $I(\cdot, \cdot)$ denotes the mutual information. Since $Z_{\mathbb{Q}}$ is independent of function $g$ and if $\sigma_Z^2 \to 0$, then $I(Z_{\mathbb{Q}} + K, K) \to 0$ (see Appendix B.2 for a proof), the contribution of the $I(Z_{\mathbb{Q}} + K, K)$ term, with respect to training $g$, becomes negligible in the case of small $\sigma_Z^2$.

Unfortunately, equation 13 is no longer valid when $\mathbb{P}_d$ lies on a manifold since $Z_{\mathbb{Q}}$ will be a singular random variable and the differential entropy $H(Z_{\mathbb{Q}})$ is not defined. In Appendix B.3, we show that leaving out the entropy $H(Z_{\tilde{\mathbb{Q}}})$ corresponds to minimizing an *upper bound* of the spread KL divergence. To further find out the contribution of the negative entropy term, we compare between leaving out $-H(Z_{\tilde{\mathbb{Q}}})$ and approximating $-H(Z_{\tilde{\mathbb{Q}}})$ during training. In Appendix B.4, we discuss the approximation technique and give the empirical evidence which shows that the ignoring $-H(Z_{\tilde{\mathbb{Q}}})$ will not affect the training of our model. Therefore, we make use of volume preserving $g$ and small variance $\sigma_Z^2 = 1 \times 10^{-4}$ in our experiments.

In contrast to other volume preserving flows, that utilize a fixed prior $\mathbb{P}_Z$, our method affords additional flexibility by way of allowing for changes to the 'volume' of the prior towards matching the distribution of the target data. In this way, our decision to employ volume-preserving flow functions does not limit the expressive power or abilities of the model, in principle. Popular non-volume preserving flow structures, e.g. affine coupling flow, may also easily be normalized to become volume preserving, thus further extending the applicability of our approach (see Appendix C.1 for an example).

**Term 2:** The noisy prior $p(\tilde{\mathbf{z}})$ is defined to be a degenerate Gaussian $\mathcal{N}(0, AA^T)$, convolved with a Gaussian noise $\mathcal{N}(0, \sigma_Z^2 I)$, and has a closed form density

$$p(\tilde{\mathbf{z}}) = \mathcal{N}(0, AA^T + \sigma_Z^2 I). \tag{14}$$

Therefore, the log density $\log p(\tilde{\mathbf{z}})$ is well defined, we can approximate term 2 by Monte Carlo

$$\int q(\tilde{\mathbf{z}}) \log p(\tilde{\mathbf{z}}) d\tilde{\mathbf{z}} \approx \frac{1}{N} \sum_{n=1}^{N} \log p(\tilde{\mathbf{z}}_n), \tag{15}$$

where $q(\tilde{\mathbf{z}}) = \int p(\tilde{\mathbf{z}}|\mathbf{z}) d\mathbb{Q}_Z$. To sample from $q(\tilde{\mathbf{z}})$, we first get a data sample $\mathbf{x} \sim \mathbb{P}_d$, use function $g$ to get $\mathbf{z} = g(\mathbf{x})$ (so $\mathbf{z}$ is a sample of $\mathbb{Q}_Z$) and finally sample $\tilde{\mathbf{z}} \sim p(\tilde{\mathbf{z}}|\mathbf{z})$.

## 7 EXPERIMENTS

To demonstrate the effectiveness of our approach, Sections 7.1–7.4 report experiments on four datasets; toy 2D and 3D data, the *fading square* dataset and a synthesized adaptation of MNIST. We use the Adam optimizer (Kingma and Ba, 2014) in all our experiments. Our flow networks are consisted of incompressible affine coupling layer introduced by Sorrenson et al. (2020); Dinh et al. (2016). We compare a volume preserving flow with a learnable prior (our method) to a non-volume preserving flow with fixed prior, so both models have the ability to adapt their 'volume' to fit the target distribution and retains fair comparisons. See Appendix C.1 for a detailed discussion of the incompressible affine coupling layer and the network structures we use for all the experiments.

### 7.1 2D TOY DATA

We firstly verify our method using the toy dataset described in Section 2 and Figure 1a. The flow function has two coupling layers. We train our model using learning rate $3 \times 10^{-4}$ and batch size 100 for 10k iterations. Figure 3 shows the samples from the model, the learned prior and the eigenvalues of $AA^T$. We observe the sample quality is better than that in Figure 1b and the prior has learned

a degenerate Gaussian with $\text{Indim}(\mathbb{P}_Z) = 1$, which matches $\text{Indim}(\mathbb{P}_d)$. In Appendix C.2, we show that our model can not only learn the manifold support of target distribution but also capture the 'density' allocation on the manifold.

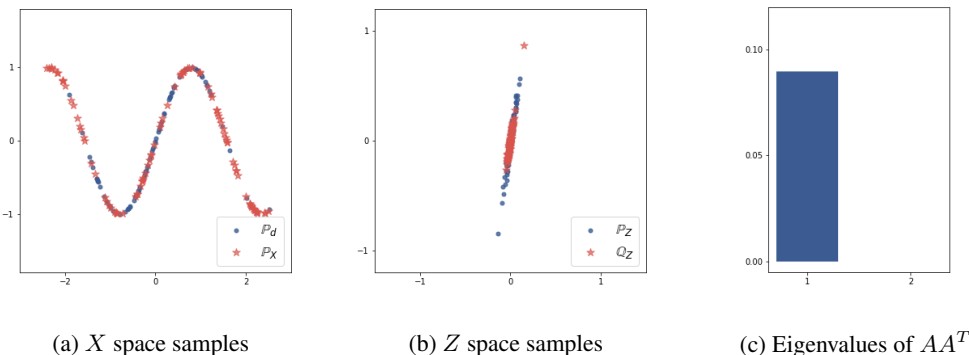

(a) $X$ space samples  (b) $Z$ space samples  (c) Eigenvalues of $AA^T$

Figure 3: (a) shows the samples from the data distribution $\mathbb{P}_d$ and our model $\mathbb{P}_X$. (b) shows the sample from the learned prior $\mathbb{P}_Z$ and the distribution $\mathbb{Q}_Z$. (c) shows the eigenvalues of $AA^T$.

## 7.2 S-CURVE DATASET

We fit our model to the S-curve dataset shown in Figure 4a. The data distribution lies on a 2D manifold in a 3D space, therefore $\text{Indim}(\mathbb{P}_d) = 2$. Specific network structure and training details can be found in Appendix C.1. After training, our model learns a nonlinear function $g$ to transform $\mathbb{P}_d$ to $\mathbb{Q}_Z$, where the latter lies on a 2D linear subspace in 3D space (see Figure 4b). Following this, a linear dimensionality reduction can be conducted to generate 2D data representations, we now briefly outline a general procedure for this.

For $\mathbb{Q}_Z$ with $\text{Amdim}(\mathbb{Q}_Z) = D$ and $\text{Indim}(\mathbb{Q}_Z) = K$, we first find the eigenvectors $\mathbf{e}^1, \ldots, \mathbf{e}^D$ of $AA^T$, sorted by their eigenvalues. When $\text{Rank}(AA^T) = K \leq D$, there exist $K$ eigenvectors with positive eigenvalues. We select the first $K$ eigenvectors and form the matrix $\mathbf{E} = [\mathbf{e}^1, \ldots, \mathbf{e}^K]$ with dimension $D \times K$. We then transform each data sample $\mathbf{x} \in \mathbb{R}^D$ into $Z$ space: $\mathbf{z} = g(\mathbf{x})$, such that $\mathbf{z} \in \mathbb{R}^D$. Afterwards, a linear projection is carried out $\mathbf{z}^{proj} = \mathbf{z}\mathbf{E}$ to obtain the lower dimensional representation $\mathbf{z}^{proj} \in \mathbb{R}^K$. This procedure can be seen as a nonlinear dimensionality reduction, where the nonlinear component is solved using the learned function $g$.

We plot the resulting representations in Figure 4b. The colormap indicates correspondence between the data in 3D space and the representation in 2D space. We observe that our method can successfully (1) identify that the intrinsic dimension of the data is two and (2) project the data into a 2D space that faithfully preserves the structure of the original data distribution. We also compare the sample generation quality with a flow that has a fixed Gaussian prior, see Appendix C.3 for details.

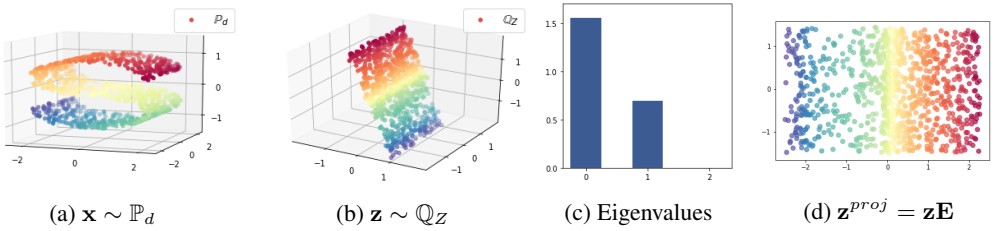

(a) $\mathbf{x} \sim \mathbb{P}_d$  (b) $\mathbf{z} \sim \mathbb{Q}_Z$  (c) Eigenvalues  (d) $\mathbf{z}^{proj} = \mathbf{z}\mathbf{E}$

Figure 4: (a) S-curve data samples $\mathbf{x} \sim \mathbb{P}_d$. (b) The latent representation $\mathbf{z} = g(\mathbf{x})$, points can be observed to lie on a linear subspace. (c) Eigenvalues of the matrix $AA^T$, we deduce that $\text{Indim}(\mathbb{P}_d) = 2$. (d) Our representation after the dimensionality reduction $\mathbf{z}^{proj} = \mathbf{z}\mathbf{E}$.

### 7.3 FADING SQUARE DATASET

The *fading square* dataset (Rubenstein et al., 2018) was proposed in order to diagnose model behavior when data distribution and model possess differing intrinsic dimension and therefore constitutes a further relevant test bed for our current work. The dataset consists of $32 \times 32$ pixel images with $6 \times 6$ grey squares on a black background. The grey scale values are sampled from a uniform distribution with range $[0, 1]$, so $\texttt{Indim}(\mathbb{P}_d) = 1$. Figure 5a shows the data samples. We fit our model to the dataset, the network structure and the training details can be found in Appendix C.

Figure 5b shows samples from our trained model. Figure 5d shows the first 20 eigenvalues of the $AA^T$ (ranked from high to low), we can see only one eigenvalue is larger than zero and the others have converged to zero. This illustrates that we have successfully identified the intrinsic dimension of $\mathbb{P}_d$. We further carry out the dimensionality reduction process that was introduced in Section 7.2; the latent representation $\mathbf{z}$ is projected onto a 1D line and we visualize the correspondence between the projected representation and the data in Figure 5e. Pixel grey-scale values can be observed to decay as the 1D representation is traversed from left to right, indicating that our representations are consistent with the properties of the original data distribution. In contrast, we find that the traditional flow model, with a fixed 1024D Gaussian $p(\mathbf{z})$, fails to learn such a data distribution, see Figure 5c.

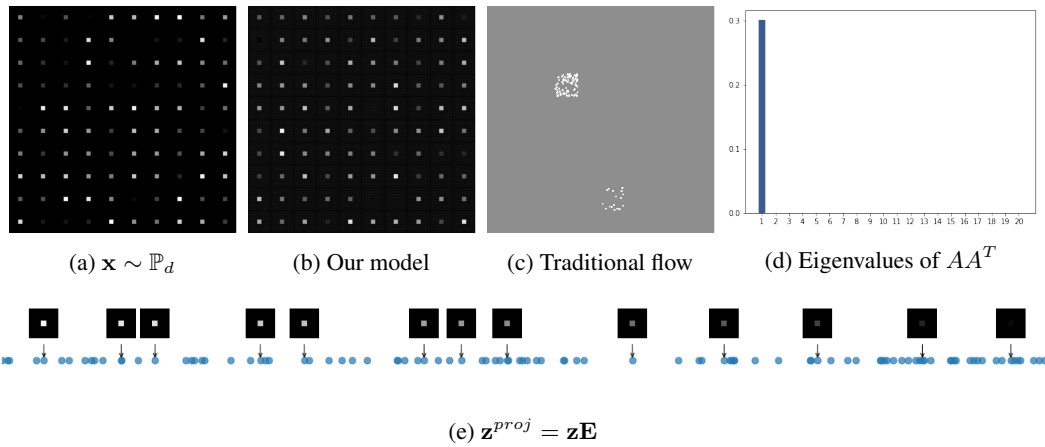

(a) $\mathbf{x} \sim \mathbb{P}_d$      (b) Our model      (c) Traditional flow      (d) Eigenvalues of $AA^T$

(e) $\mathbf{z}^{proj} = \mathbf{z}\mathbf{E}$

Figure 5: (a) and (b) show the samples from the data distribution and our model, respectively. (c) shows a traditional flow based model with a fixed Gaussian prior fails to generate any valid samples. (d) shows the first 20 eigenvalues of the matrix $AA^T$. (e) visualization of the representation after applying dimensionality reduction. See text for further discussion.

### 7.4 MNIST DATA

We further investigate training of our model using images of digits. However, for datasets like MNIST (LeCun, 1998), the true intrinsic dimension is unknown. In order to verify the correctness of our model's ability to identify the intrinsic data dimension, we construct synthetic datasets by first fitting an implicit model $p_\theta(\mathbf{x}) = \int \delta(\mathbf{x} - g(\mathbf{z})p(\mathbf{z}))d\mathbf{z}$ to the MNIST dataset, and then use samples from the trained model $\mathbf{x} \sim p_\theta(\mathbf{x})$ as training data. The intrinsic dimension of the training dataset is the same as the dimension of the latent variable $\mathbf{z}$ in the implicit model $\texttt{Indim}(\mathbb{P}_d) = \dim(\mathbf{z})$. We construct two datasets with $\dim(\mathbf{z}) = 5$ and $\dim(\mathbf{z}) = 10$ such that $\texttt{Indim}(\mathbb{P}_d) = 5$ and $\texttt{Indim}(\mathbb{P}_d) = 10$, respectively. Further details on the implicit model, flow network structure, training method and samples from the learned models are found in Appendix C.

In contrast to the fading square dataset, we find that in order to successfully train the model (*i.e.* such that valid image samples are generated), it is *necessary* to add small Gaussian noise to the training data. This trick is commonly used in the training of flow based models for image data (Sorrenson et al., 2020). We note that adding Gaussian noise breaks the assumption that the data lies on a manifold and, alternatively, the intrinsic dimension of the training data distribution will be equal to its ambient dimension. Towards alleviating this undesired effect, we firstly add Gaussian noise with standard deviation $\sigma_x = 0.05$ and anneal $\sigma_x$ after $2000k$ iterations with a factor of 0.9 every $10k$

iterations. However, to help retain successful model training, we disallow the annealing procedure to reach a state where zero noise is added due to the outlined model behavior observed when considering this image dataset. Experimentally, we cap a lower-bound Gaussian noise level of 0.01 and leave further investigation of this phenomenon to future work.

In Figure 6a and 6b, we plot the first 20 eigenvalues of the $AA^T$ (ranked from high to low) after training on two synthetic MNIST datasets with intrinsic dimension 5 and 10. It can be observed that 5 and 10 eigenvalues are significantly larger than the remaining values, respectively. It can thus be concluded that the intrinsic dimension of the two datasets are 5 and 10. Remaining non-zero eigenvalues can be attributed to the small Gaussian noise added to the training data.

For the original MNIST dataset, it was shown that digits have different intrinsic dimension (Costa and Hero, 2006). This suggests the distribution of MNIST may lie on *several, disconnected* manifolds with differing intrinsic dimension. Although our model assumes that $\mathbb{P}_d$ lies on one continuous manifold, it is interesting to investigate the case when this model assumption is not fulfilled. We thus fit our model to the original MNIST data and plot the eigenvalues in Figure 6c. In contrast to Figures 6a and 6b, the gap between eigenvalues can be seen to be less pronounced, with no obvious step change. However the values suggest that the intrinsic dimension of MNIST is between 11 and 14. This result is consistent with previous estimations stating that the intrinsic dimension of MNIST is between 12 and 14 (Facco et al., 2017; Hein and Audibert, 2005; Costa and Hero, 2006). Recent work Cornish et al. (2019) discusses fitting flow models to a $\mathbb{P}_d$ that lies on disconnected components, by introducing a mixing prior. Such techniques may be easily combined with our method towards constructing more powerful flow models; a promising direction for future work.

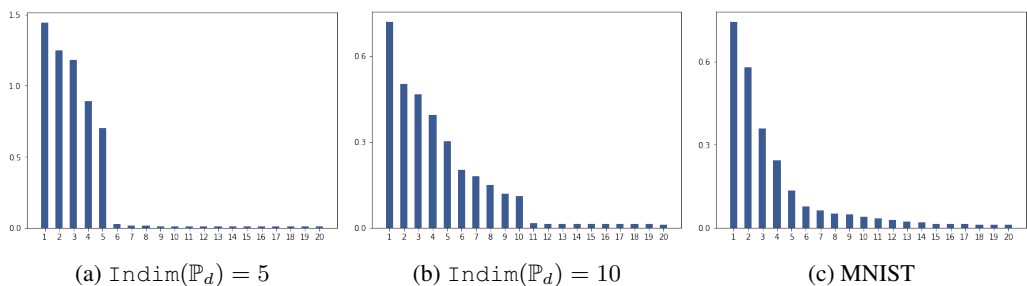

(a) Indim$(\mathbb{P}_d) = 5$  (b) Indim$(\mathbb{P}_d) = 10$  (c) MNIST

Figure 6: This figure shows the eigenvalues of the $AA^T$ after fitting the (a) synthetic MNIST with intrinsic dimension 5; (b) synthetic MNIST with intrinsic dimension 10; (c) original MNIST dataset.

## 8 RELATED WORK

Classic latent variable generative models assume that data distributions lie *around* a low-dimensional manifold, for example the Variational Auto-Encoder (Kingma and Welling, 2013) or, recently introduced, Noisy Injective Flows (Cunningham et al., 2020). Such methods typically assume that observational noise is not degenerated (e.g. a fixed Gaussian distribution), therefore the model distribution is absolutely continuous and maximum likelihood learning is thus well defined. However, common distributions such as natural images usually don't have Gaussian observational noise (Zhao et al., 2017). Therefore, in this work, we focus on modeling distributions that lie *on* a low-dimensional manifold.

The study of manifold learning for nonlinear dimensionality reduction Cayton (2005) and intrinsic dimension estimation Camastra and Staiano (2016) is a rich field with an extensive set of tools. However, most methods commonly do not model data density on the manifold and are thus not used for the same purpose as the models introduced here. There are however a number of recent works that introduced normalizing flows on manifolds that we now highlight and relate to our approach.

Several works define flows on manifolds with prescribed charts. Gemici et al. (2016) generalized flows from Euclidean spaces to Riemannian manifolds by proposing to map points from the manifold $\mathcal{M}$ to $\mathbb{R}^K$, apply a normalizing flow in this space and then map back to $\mathcal{M}$. The technique has been

further extended to Tori and Spheres (Rezende et al., 2020). These methods require knowledge of the intrinsic dimension $K$ and a parameterization of the coordinate chart of the data manifold.

Without providing a chart mapping a priori, M-flow (Brehmer and Cranmer, 2020) recently proposed an algorithm that learns the chart mapping and distribution density simultaneously. However, their method still requires that the dimensionality of the manifold is known. They propose that the dimensionality can be learned either by a brute-force solution or through a trainable variance in the density function. The brute-force solution is clearly infeasible for data embedded in extremely high dimensional space, as is often encountered in deep learning tasks. Use of a trainable variance is natural and similar to our approach. However, as discussed at the beginning of this paper, without carefully handling the KL or MLE term in the objective, a vanishing variance parameter will result in wild behavior of the optimization process since these terms are not well defined.

While the GIN model considered in Sorrenson et al. (2020) could recover the low dimensional generating latent variables following their identifiability theorem, the assumptions therein require knowledge of an auxiliary variable, *e.g.* the label, which is not required in our model. Behind this superficial difference is the essential discrepancy between the concept of informative dimensions and intrinsic dimensions. The GIN model discovers the latent variables that are informative in a given context, defined by the auxiliary variable $u$ instead of the true intrinsic dimensions. In their synthetic example, the ten dimensional data is a nonlinear transformation of ten dimensional latent variables where two out of ten are correlated with the labels of the data and the other eight are not. In this example, there are two informative dimensions, but there are ten intrinsic dimensions. Nevertheless, our method for intrinsic dimension discovery can be used together with informative dimension discovery methods to discover finer structures of data.

## 9 CONCLUSION

We presented a principled strategy to learn the data distribution that lies on a manifold and identify its intrinsic dimension. We fix the ill-defined KL divergence and show, across multiple datasets, the resulting benefits for flow based models under both sample generation and representation quality. There remain a number of open questions and interesting directions for future work. Namely; further exploration of the effects of the entropy term in the case of non-volume preserving networks and, additionally, investigation of the phenomenon concerning the apparent necessity of noise addition in cases pertaining to complex real-world distributions *e.g.* image data.

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

## A  MAXIMUM LIKELIHOOD ESTIMATION AND KL DIVERGENCE

Given data $x_1, x_2, \ldots, x_N$ sampled independently from the true data distribution $\mathbb{P}_d$, with density function $p_d(x)$[3], we want to fit the model density $p(x)$[3] to the data. A popular choice to achieve this involves minimization of the KL divergence where:

$$\text{KL}(p_d(x)||p(x)) = \int p_d(x) \log p_d(x) dx - \int p_d(x) \log p(x) dx \tag{16}$$

$$= - \int p_d(x) \left( \log p_Z(g(x)) + \log \left| \det \left( \frac{\partial g}{\partial x} \right) \right| \right) dx + const.. \tag{17}$$

Since we can only access samples from $p_d(x)$, we approximate the integral by Monte Carlo sampling

$$\text{KL}(p_d(x)||p(x)) \approx - \frac{1}{N} \sum_{n=1}^{N} \log p(x_n) + const.. \tag{18}$$

Therefore, minimizing the KL divergence between the data distribution and the model is (approximately) equivalent to Maximum Likelihood Estimation (MLE).

When $p(x)$ is a flow-based model with invertible flow function $f : Z \to X, g = f^{-1}$, minimizing the KL divergence in $X$ space is equivalent to minimizing the KL divergence in the $Z$ space. We let $X_d$ be the random variable of data distribution and define $Z_{\mathbb{Q}} : Z_{\mathbb{Q}} = g(X_d)$ with density $q(z)$, so $q(z)$ can be represented as

$$q(z) = \int \delta(z - g(x)) p_d(x) dx. \tag{19}$$

Let $p(z)$ be the density of the prior distribution $\mathbb{P}_Z$, the KL divergence in $Z$ space can be written as

$$\text{KL}(q(z)||p(z)) = \underbrace{\int q(z) \log q(z) dz}_{\text{Term 1}} - \underbrace{\int q(z) \log p(z) dz}_{\text{Term 2}} . \tag{20}$$

Term 1: using the properties of transformation of random variable (Papoulis and Pillai, 2002, pp. 660), the negative entropy can be written as

$$\int q(z) \log q(z) dz = \underbrace{\int p_d(x) \log p_d(x) dx}_{const.} - \int p_d(x) \log \left| \det \left( \frac{\partial g}{\partial x} \right) \right| dx. \tag{21}$$

Term2: the cross entropy can be written as

$$\int q(z) \log p(z) dz = \int \int \delta(z - g(x)) p_d(x) \log p(z) dz dx \tag{22}$$

$$= \int p_d(x) \log p(g(x)) dx. \tag{23}$$

Therefore, the KL divergence in $Z$ space is equivalent to the KL divergence in $X$ space

$$\text{KL}(q(z)||p(z)) = \text{KL}(p_d(x)||p(x)). \tag{24}$$

We thus build the connection between MLE and minimizing the KL divergence in $Z$ space.

## B  ENTROPY

### B.1  AN EXAMPLE

Assume a 2D Gaussian random variable $X$ with covariance $\begin{bmatrix} 1 & 0 \\ 0 & 1 \end{bmatrix}$, $g$ is a volume preserving flow with parameter $\theta$. Let $Z_1 = g_{\theta_1}(Z)$ be a Gaussian with covariance $\begin{bmatrix} 2 & 0 \\ 0 & \frac{1}{2} \end{bmatrix}$ and $Z_2 = g_{\theta_2}(X)$ be

---

[3]For simplicity, we use notation $p(x)$ to represent the model $p_X(x)$ unless otherwise specified.

a Gaussian with covariance $\begin{bmatrix} 3 & 0 \\ 0 & \frac{1}{3} \end{bmatrix}$. Therefore the entropy $\mathrm{H}(Z_1) = \mathrm{H}(Z_2) = \mathrm{H}(X)$ and doesn't

not depend on $\theta$. Let $K$ be an Gaussian with zero mean and covariance $\begin{bmatrix} 1 & 0 \\ 0 & 1 \end{bmatrix}$, so $Z_1 + K$ is a

Gaussian with covariance $\begin{bmatrix} 3 & 0 \\ 0 & \frac{3}{2} \end{bmatrix}$ and $Z_2 + K$ is a Gaussian with covariance $\begin{bmatrix} 4 & 0 \\ 0 & \frac{4}{3} \end{bmatrix}$. Therefore

$\mathrm{H}(Z_1 + K) \neq \mathrm{H}(Z_2 + K)$ and $\mathrm{H}(g_\theta(X) + K)$ depends on $\theta$. A similar example can be constructed when $X$ is not absolutely continuous.

## B.2   $Z$ IS AN ABSOLUTELY CONTINUOUS RANDOM VARIABLE

For two mutually independent absolutely continuous random variable $Z$ and $K$, the mutual information between $Z + K$ and $K$ is

$$\mathrm{I}(Z + K, K) = \mathrm{H}(Z + K) - \mathrm{H}(Z + K|K) \tag{25}$$
$$= \mathrm{H}(Z + K) - \mathrm{H}(Z|K) \tag{26}$$
$$= \mathrm{H}(Z + K) - \mathrm{H}(Z). \tag{27}$$

The last equality holds because $Z$ and $K$ are independent. Since mutual information $\mathrm{I}(Z+K, K) \geq 0$, we have

$$\mathrm{H}(Z) \leq \mathrm{H}(Z + K) = \mathrm{H}(Z) + \mathrm{I}(Z + K, K). \tag{28}$$

Assume $K$ has a Gaussian distribution with 0 mean and variance $\sigma_Z^2$[4]. When $\sigma_Z^2 = 0$, $K$ degenerates to a delta function, so $Z + K = Z$ and

$$\mathrm{I}(Z + K, K) = \mathrm{H}(Z + K) - \mathrm{H}(Z) = 0, \tag{29}$$

this is because the mutual information between an a.c. random variable and a singular random variable is still well defined, see (Yeung, 2008, Theorem 10.33). Assume $K_1$, $K_2$ are Gaussian random variables with 0 mean and variances $\sigma_1^2$ and $\sigma_2^2$ respectively. Without loss of generality, we assume $\sigma_1^2 > \sigma_2^2$ and $\sigma_1^2 = \sigma_2^2 + \sigma_\delta^2$, and let $K_\delta$ be the random variable of a Gaussian that has 0 mean and variance $\sigma_\delta^2$ such that $K_1 = K_2 + K_\delta$. By the data-processing inequality, we have

$$\mathrm{I}(Z + K_2, K_2) \leq \mathrm{I}(Z + K_2 + K_\delta, K_2 + K_\delta) = \mathrm{I}(Z + K_1, K_1). \tag{30}$$

Therefore, $\mathrm{I}(Z + K, K)$ is a monotonically decreasing function when $\sigma_Z^2$ decreases and when $\sigma_Z^2 \to 0$, $I(Z + K, K) \to 0$.

## B.3   UPPER BOUND OF THE SPREAD KL DIVERGENCE

In this section, we show that leaving out the entropy term $\mathrm{H}(Z_{\tilde{\mathbb{Q}}})$ in equation 12 is equivalent to minimizing an *upper bound* of the spread KL divergence.

For singular random variable $Z_{\mathbb{Q}} = g(X_d)$ and absolutely continuous random variable $K$ that are independent, we have

$$\mathrm{H}(Z_{\mathbb{Q}} + K) - \mathrm{H}(K) = \mathrm{H}(Z_{\mathbb{Q}} + K) - \mathrm{H}(Z_{\mathbb{Q}} + K|Z_{\mathbb{Q}}) \tag{31}$$
$$= \mathrm{I}(Z_{\mathbb{Q}} + K, Z_{\mathbb{Q}}) \geq 0. \tag{32}$$

The second equation is from the definition of Mutual Information (MI); the MI between an a.c. random variable and a singular random variable is well defined and always positive, see (Yeung, 2008, Theorem 10.33) for a proof.

Therefore, we can construct an upper bound of the spread KL objective in equation 12

$$\mathrm{KL}(\tilde{q}||\tilde{p}) = \underbrace{\int q(\tilde{\mathbf{z}}) \log q(\tilde{\mathbf{z}}) d\tilde{\mathbf{z}}}_{-\mathrm{H}(Z_{\mathbb{Q}}+K)} - \int q(\tilde{\mathbf{z}}) \log p(\tilde{\mathbf{z}}) d\tilde{\mathbf{z}} \tag{33}$$

$$\leq \underbrace{-\mathrm{H}(K)}_{const.} - \int q(\tilde{\mathbf{z}}) \log p(\tilde{\mathbf{z}}) d\tilde{\mathbf{z}}. \tag{34}$$

Therefore, ignoring the negative entropy term during training is equivalent to minimizing an upper bound of the spread KL objective.

---

[4]The extension to higher dimensions is straightforward.

### B.4 EMPIRICAL EVIDENCE FOR IGNORING THE NEGATIVE ENTROPY

In this section, we first introduce the approximation technique to compute the negative entropy term, and then discuss the contribution of this term for the training. The negative entropy of random variable $Z_{\tilde{Q}}$ is

$$-\mathrm{H}(Z_{\tilde{\mathbb{Q}}}) = \int q(\tilde{\mathbf{z}}) \log q(\tilde{\mathbf{z}}) d\tilde{\mathbf{z}}, \tag{35}$$

where

$$q(\tilde{\mathbf{z}}) = \int_{\mathbf{z}} p_K(\tilde{\mathbf{z}} - \mathbf{z}) d\mathbb{Q}_Z, \tag{36}$$

and $p_K$ is the density of a Gaussian with diagonal covariance $\sigma_Z^2 I$. We first approximate $\tilde{q}(\tilde{\mathbf{z}})$ by a mixture of Gaussians

$$q(\tilde{\mathbf{z}}) \approx \frac{1}{N} \sum_{n=1}^{N} \mathcal{N}(\tilde{\mathbf{z}}; \mathbf{z}^n, \sigma_Z^2 I) \equiv \hat{q}^N(\tilde{\mathbf{z}}) \tag{37}$$

where $\mathbf{z}^n$ is the $n$th sample from distribution $\mathbb{Q}_z$ by first sampling $\mathbf{x}^n \sim \mathbb{P}_d$ and letting $\mathbf{z}^n = g(\mathbf{x}^n)$. We denote the random variable of this Gaussian mixture as $\hat{Z}_{\tilde{\mathbb{Q}}}^N$ and approximate

$$-\mathrm{H}(Z_{\tilde{\mathbb{Q}}}) \approx -\mathrm{H}(\hat{Z}_{\tilde{\mathbb{Q}}}^N). \tag{38}$$

However, the entropy of a Gaussian mixture distribution does not have a closed form, so we further conduct a first order Taylor expansion approximation (Huber et al., 2008)

$$-\mathrm{H}(\hat{Z}_{\tilde{\mathbb{Q}}}^N) \approx \frac{1}{N} \sum_{n=1}^{N} \log \hat{q}^N(\tilde{\mathbf{z}} = \mathbf{z}^n), \tag{39}$$

this approximation is accurate for small $\sigma_Z^2$. Finally we have our approximation

$$-\mathrm{H}(Z_{\tilde{\mathbb{Q}}}) \approx \frac{1}{N} \sum_{n=1}^{N} \log \hat{q}^N(\tilde{\mathbf{z}} = \mathbf{z}^n). \tag{40}$$

To evaluate the contribution of the negative entropy, we train our flow model on both low dimensional data (Toy datasets: 2D) and high dimensional data (Fading square dataset: 1024D) by optimization that uses two objectives: (1) ignoring the negative entropy term in equation 7 and (2) approximating the negative entropy term in equation 7 using the approximation discussed above. During training, we keep tracking of the value of the entropy $\mathrm{H}(Z_{\tilde{\mathbb{Q}}})$ (using the approximation value) in both objectives. We let $N$ equal the batch size when approximating the entropy. Additional training details remain consistent with those described in Appendix C.

To evaluate the contribution of the negative entropy, we train our flow model on both low dimensional data (Toy datasets: 2D) and high dimensional data (Fading square dataset: 1024D) by optimization that uses two objectives: (1) ignoring the negative entropy term in equation 7 and (2) approximating the negative entropy term in equation 7 using the approximation discussed above. During training, we keep tracking of the value of the entropy $\mathrm{H}(Z_{\tilde{\mathbb{Q}}})$ (using the approximation value) in both objectives. We let $N$ equal the batch size when approximating the entropy. Additional training details remain consistent with those described in Appendix C.

In Figure 7, we plot the (approximated) entropy value $\mathrm{H}(Z_{\tilde{\mathbb{Q}}})$ during training for both experiments. We find the difference between having the (approximated) negative entropy term and ignoring the negative entropy to be negligible. We leave theoretical investigation on the effects of leaving out the entropy term during training to future work.

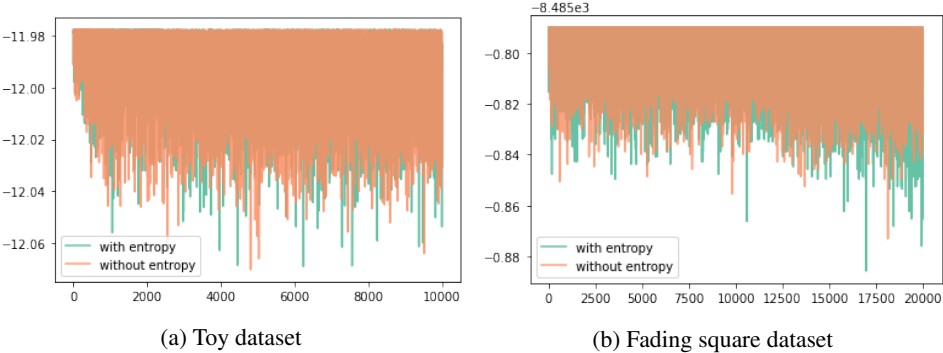

(a) Toy dataset                    (b) Fading square dataset

Figure 7: Figure (a) and (b) show the (approximated) entropy value using two different training objectives, for two different experiments.

## C  EXPERIMENTS

### C.1  NETWORK ARCHITECTURE

The flow network we use consists of incompressible affine coupling layers (Sorrenson et al., 2020; Dinh et al., 2016). Each coupling layer splits a $D$-dimensional input $\mathbf{x}$ into two parts $\mathbf{x}_{1:d}$ and $\mathbf{x}_{d+1:D}$. The output of the coupling layer is

$$\mathbf{y}_{1:d} = \mathbf{x}_{1:d} \tag{41}$$
$$\mathbf{y}_{d+1:D} = \mathbf{x}_{d+1:D} \odot \exp(s(\mathbf{x}_{1:d})) + t(\mathbf{x}_{1:d}), \tag{42}$$

where $s : \mathbb{R}^d \to \mathbb{R}^{D-d}$ and $t : \mathbb{R}^d \to \mathbb{R}^{D-d}$ are scale and translation functions parameterized by neural networks, $\odot$ is the element-wise product. The log-determinant of the Jacobian of a coupling layer is the sum of the scaling function $\sum_j s(\mathbf{x}_{1:d})_j$. To make the coupling transform volume preserving, we normalize the output of the scale function, so the $i$-th dimension of the output is

$$\tilde{s}(\mathbf{x}_{1:d})_i = s(\mathbf{x}_{1:d})_i - \frac{1}{D-d} \sum_j s(\mathbf{x}_{1:d})_j, \tag{43}$$

and the log-determinant of the Jacobian is $\sum_i \tilde{s}(\mathbf{x}_{1:d})_i = 0$. We compare a volume preserving flow with a learnable prior (normalized $s(\cdot)$) to a non-volume preserving flow with fixed prior (un-normalized $s(\cdot)$). In this way both models have the ability to adapt their 'volume' to fit the target distribution, retaining comparison fairness.

In our affine coupling layer, the scale function $s$ and the translation function $t$ have two types of structure: fully connected net and convolution net. Each fully connected network is a 4-layer neural network with hidden-size 24 and Leaky ReLU with slope 0.2. Each convolution net is a 3-layer convolutional neural network with hidden channel size 16, kernel size $3\times3$ and padding size 1. The activation is Leaky ReLU with slope 0.2. The downsampling decreases the image width and height by a factor of 2 and increases the number of channels by 4 in a checkerboard-like fashion (Sorrenson et al., 2020; Jacobsen et al., 2018). When multiple convolutional nets are connected together, we randomly permute the channels of the output of each network except the final one. In Table 1, 2, 3, 4, we show the network structures for our four main paper experiments.

| Type of block | Number | Input shape | Affine coupling layer widths |
|---|---|---|---|
| Fully connected | 2 | 2 | $1 \to 24 \to 24 \to 24 \to 1$ |

Table 1: The network structure for toy datasets.

| Type of block | Number | Input shape | Affine coupling layer widths |
|---|---|---|---|
| § Fully connected | 6 | 3 | even: $2 \to 24 \to 24 \to 24 \to 1$
odd: $1 \to 24 \to 24 \to 24 \to 2$ |

Table 2: The network structure for S-Curve dataset. If we denote the input vector of each coupling is $\mathbf{x}$, the function $s$ and $t$ takes $\mathbf{x}[0]$ in the first coupling layer and $\mathbf{x}[1]$ in the second coupling layer.

| Type of block | Number | Input shape | Affine coupling layer widths |
|---|---|---|---|
| Downsampling | 1 | $(1, 32, 32)$ | |
| Convolution | 2 | $(4, 16, 16)$ | $2 \to 16 \to 16 \to 4$ |
| Downsampling | 1 | $(4, 16, 16)$ | |
| Convolution | 2 | $(16, 8, 8)$ | $8 \to 16 \to 16 \to 16$ |
| Flattening | 1 | $(16, 8, 8)$ | |
| Fully connected | 2 | 1024 | $512 \to 24 \to 24 \to 24 \to 512$ |

Table 3: The network structure for our fading square dataset experiments. If we denote the input vector of each coupling to be $\mathbf{x}$, the functions $s$ and $t$ take values $\mathbf{x}[1{:}2]$ in the even coupling layer and $\mathbf{x}[3]$ in the odd coupling layer.

| Type of block | Number | Input shape | Affine coupling layer widths |
|---|---|---|---|
| Downsampling | 1 | $(1, 28, 28)$ | |
| Convolution | 4 | $(4, 14, 14)$ | $2 \to 16 \to 16 \to 4$ |
| Downsampling | 1 | $(4, 14, 14)$ | |
| Convolution | 4 | $(16, 7, 7)$ | $8 \to 16 \to 16 \to 16$ |
| Flattening | 1 | $(16, 7, 7)$ | |
| Fully connected | 2 | 784 | $392 \to 24 \to 24 \to 24 \to 392$ |

Table 4: Network structure for synthetic MNIST dataset experiment.

## C.2 TOY DATASET

We also construct a second dataset and train a flow model using the same training procedure discussed in Section 7.1. Figure 8a shows the samples from the data distribution $\mathbb{P}_d$, each data point is a 2D vector $\mathbf{x} = [x_1, x_2]$ where $x_1 \sim \mathcal{N}(0, 1)$ and $x_2 = x_1$, so $\texttt{Indim}(\mathbb{P}_d) = 1$. Figure 8f shows that the prior $\mathbb{P}_Z$ has learned the true intrinsic dimension $\texttt{Indim}(\mathbb{P}_Z) = \texttt{Indim}(\mathbb{P}_d) = 1$. We compare to samples drawn from a flow model that uses a fixed 2D Gaussian prior, with results shown in Figure 8. We can observe, for this simple dataset, flow with a fix Gaussian prior can generate reasonable samples, but the 'curling up' behavior, discussed in the main paper, remains highly evident in the $Z$ space, see Figure 8c.

We also plot the 'density allocation' on the manifold for the two toy datasets. For example, for the data generation process $\mathbf{x} = [x_1, x_2]$ where $x_1 \sim p = \mathcal{N}(0, 1)$ and $x_2 = x_1$, we use the density value $p(x = x_1)$ to indicate the 'density allocation' on the 1D manifold. To plot the 'density allocation' of our learned model, we first sample $\mathbf{x}_s$ uniformly from the support of the data distribution, the subscript 's' here means that they only contain the information of the *support*. Specifically, since $\mathbf{x}_s = [x_1^s, x_2^s]$, we sample $x_1^s \sim p = \mathcal{U}(-3\sigma, 3\sigma)$ ($\sigma$ is the standard deviation of $\mathcal{N}(0, 1)$, $\mathcal{U}$ is the uniform distribution) and let $x_2^s = x_1^s$ or $x_2^s = \sin(x_1^s)$, depending on which dataset is used. We use the projection procedure that was described in Section 7.2 to obtain the projected samples $\mathbf{z}_s^{proj}$, so $\mathbf{z}_s^{proj} \in \mathbb{R}^{\texttt{Indim}(\mathbb{P}_d)}$. We also project the learned prior $\mathbb{P}_Z$ to $\mathbb{R}^{\texttt{Indim}(\mathbb{P}_d)}$ by constructing $\mathbb{P}_Z^{proj}$ as a Gaussian with zero mean and a diagonal covariance contains the non-zeros eigenvalues of $AA^T$. Therefore $\mathbb{P}_Z^{proj}$ is a.s. in $\mathbb{R}^{\texttt{Indim}(\mathbb{P}_d)}$ and we denote its density function as $p^{proj}(\mathbf{z})$. We then use the density value $p^{proj}(\mathbf{z} = \mathbf{z}_s^{proj})$ to indicate the 'density allocation' at the location of $\mathbf{x}_s$ on the manifold support. In Figure 9, we compare our model with the ground truth and find that we can successfully capture the 'density allocation' on the manifold.

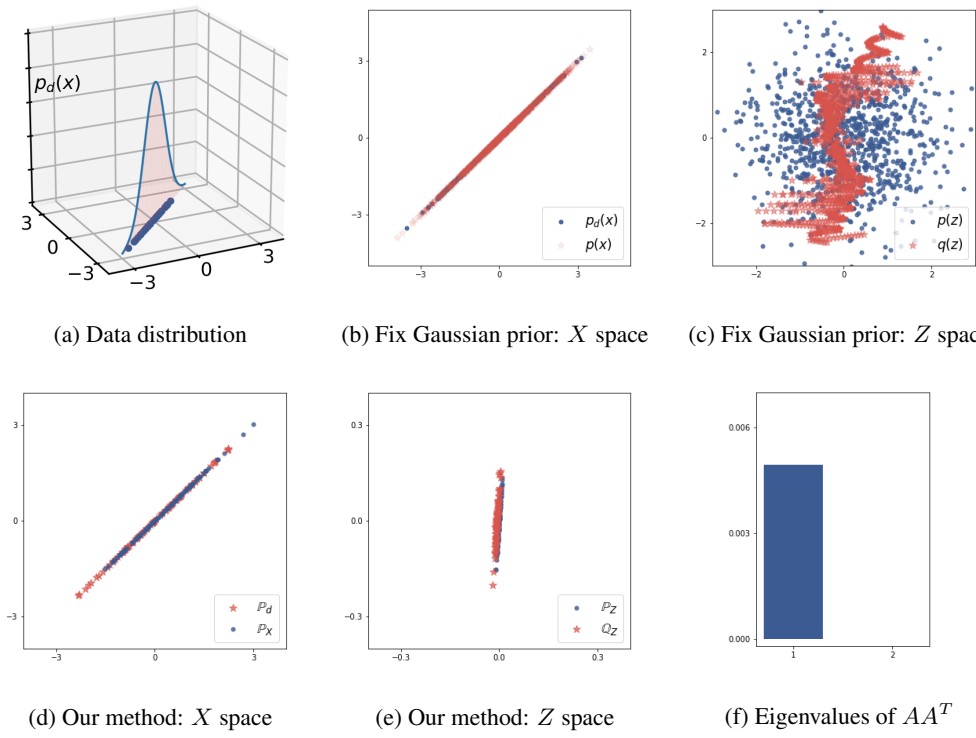

(a) Data distribution     (b) Fix Gaussian prior: $X$ space     (c) Fix Gaussian prior: $Z$ space

(d) Our method: $X$ space     (e) Our method: $Z$ space     (f) Eigenvalues of $AA^T$

Figure 8: (a) shows the samples from the data distribution. (b) and (d) show samples from a flow with a fixed Gaussian prior and our method, respectively. (c) and (d) show the latent space in both models. In (f), we plot the eigenvalues of the matrix $AA^T$.

### C.3 S-Curve dataset

To fit our model to the data, we use the Adam optimizer with learning rate $5 \times 10^{-4}$ and batch size $500$ and train the model for $200k$ iterations. We anneal the learning rate with a factor of $0.9$ every $10k$ iterations.

We compare our method with a traditional normalizing flow with a fixed 3D Gaussian prior. Both models have the same network architecture and training procedure. Figure 10a and 10b show the samples form our model and the traditional flow with a fixed 3D Gaussian prior. We can observe more samples lying outwith the true data distribution in Figure 10b than in Figure 10a. We can conclude that our model has better sample quality considering this S-Curve dataset. We also compare the latent representation for both models, see Figure 10c and 10d. We can see the representation distribution $\mathbb{Q}_Z$ captures the structure of the data distribution well whereas the distribution $\mathbb{Q}_Z$ in Figure 10d is unable to do so.

### C.4 Fading Square dataset

To fit the data, we train our model for $20k$ iterations with batch-size $100$ using the Adam optimizer. The learning rate is initialized to $5\times10^{-4}$ and decays with a factor of $0.9$ at every $1k$ iterations. We additionally use an $L2$ weight decay with factor $0.1$.

### C.5 MNIST Dataset

#### C.5.1 Implicit data generation model

To fit an implicit model to the MNIST dataset, we first train a Variational Auto-Encoder (VAE) (Kingma and Welling (2013)) with Gaussian prior $p(\mathbf{z})=\mathcal{N}(0,I)$. The encoder is $q(\mathbf{z}|\mathbf{x}) = \mathcal{N}(\mu_\theta(\mathbf{x}), \Sigma_\theta(\mathbf{x}))$ where $\Sigma$ is a diagonal matrix. Both $\mu_\theta$ and $\Sigma_\theta$ are parameterized by a 3-layer

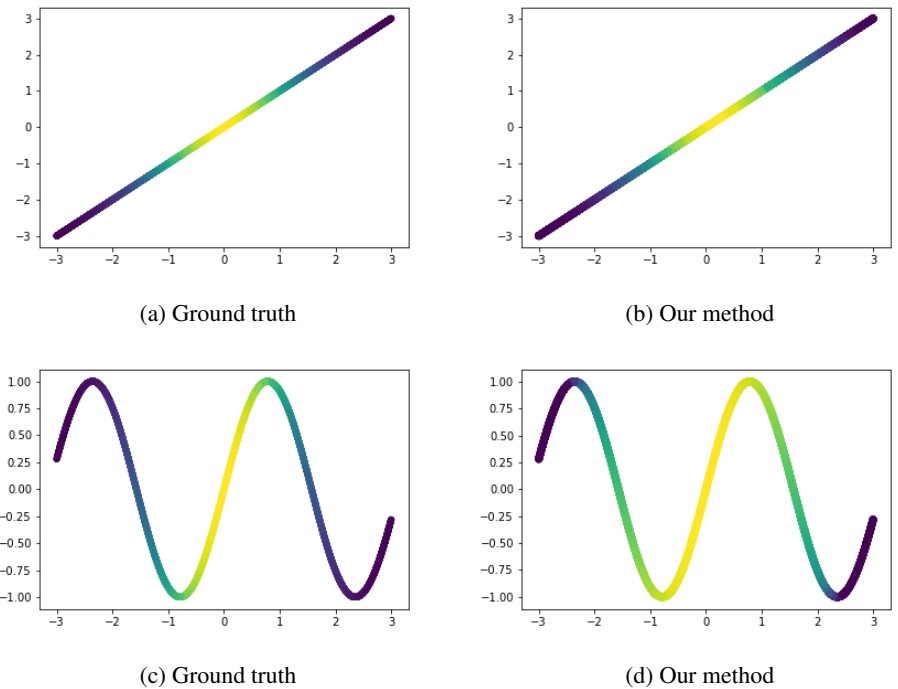

Figure 9: (a) and (c) shows the ground truth 'density allocation' on the manifold for two toy datasets, (b) and (d) shows the 'density allocation' learned by our models.

feed-forward neural network with a ReLU activation and the size of the two hidden outputs are 400 and 200. We use a Gaussian decoder $p(\mathbf{x}|\mathbf{z}) = \mathcal{N}(g_\theta(\mathbf{z}), \sigma_\mathbf{x}^2 I)$ with fixed variance $\sigma_\mathbf{x} = 0.3$. The $g_\theta$ is parameterized by a 3-layer feed-forward neural network with hidden layer sizes 200 and 400. The activation of the hidden output uses a ReLU and we utilize a Sigmoid function in the final layer output to constrain the output between 0 and 1. The training objective is to maximize the lower bound of the log likelihood

$$\log p(\mathbf{x}) \geq \int q(\mathbf{z}|\mathbf{x}) \log p(\mathbf{x}|\mathbf{z}) dz - \mathrm{KL}(q(\mathbf{z}|\mathbf{x})||p(\mathbf{z})),$$

see Kingma and Welling (2013) for further details. We use an Adam optimizer with learning rate $1 \times 10^{-4}$ and batch size 100 to train the model for 100 epochs. After training, we sample from the model by first taking a sample $\mathbf{z} \sim p(\mathbf{z})$ and letting $\mathbf{x} = g_\theta(\mathbf{z})$. This is equivalent to taking a sample from an implicit model $p_\theta(\mathbf{x}) = \int \delta(\mathbf{x} - g_\theta(\mathbf{z})) d(\mathbf{z}) d\mathbf{z}$, see Tolstikhin et al. (2017) for further discussion regarding this implicit model construction. In Figure 11, we plot samples from the trained implicit model with $\mathrm{dim}(\mathbf{z}) = 5$, $\mathrm{dim}(\mathbf{z}) = 10$ and the original MNIST data.

### C.5.2 FLOW MODEL TRAINING

We train our flow models to fit the synthetic MNIST dataset with intrinsic dimensions 5 and 10 and the original MNIST dataset. In all models, we use the Adam optimizer with learning rate $5 \times 10^{-4}$ and batch size 100. We train the model for $3000k$ iterations and, following the initial $1000k$ iterations, the learning rate decays every $10k$ iterations by a factor 0.9. In Figure 12, we plot the samples from our models trained on three different training datasets. In Figure 13, we also plot the samples from traditional flow models that trained on these three datasets using the same experiment settings , we found the samples from our models are sharper than that from traditional flow models.

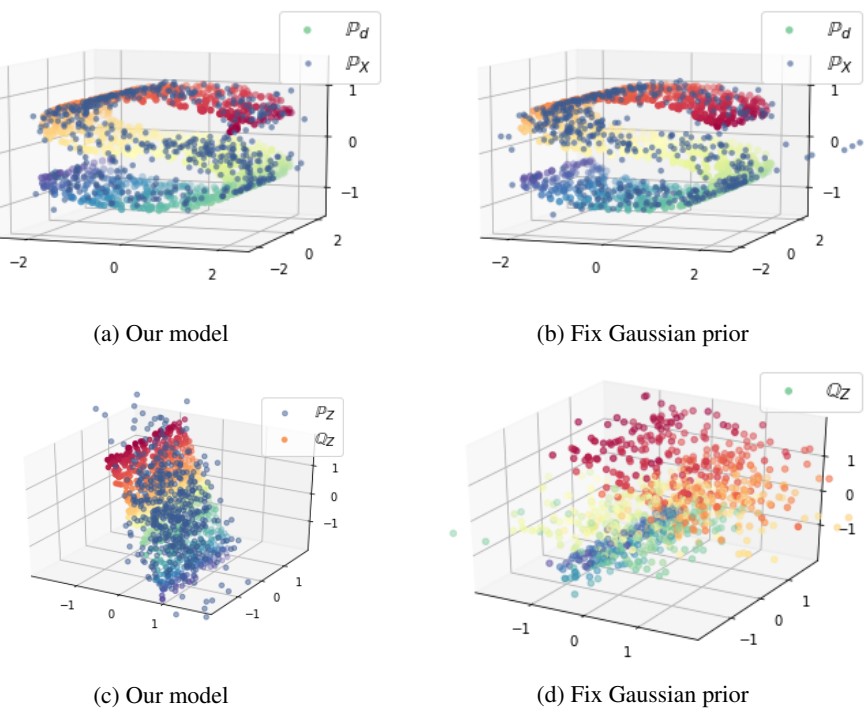

(a) Our model

(b) Fix Gaussian prior

(c) Our model

(d) Fix Gaussian prior

Figure 10: Figure (a) and (b) plot the data distribution $\mathbb{P}_Z$ of the S-Curve dataset and the samples from our model and a traditional flow with a fixed Gaussian prior. Figure (c) shows the representation distribution $\mathbb{Q}_Z$ and the learned prior $\mathbb{P}_Z$ of our model. In Figure (d), we plot the representation distribution $\mathbb{Q}_Z$ using the flow with a fixed Gaussian prior.

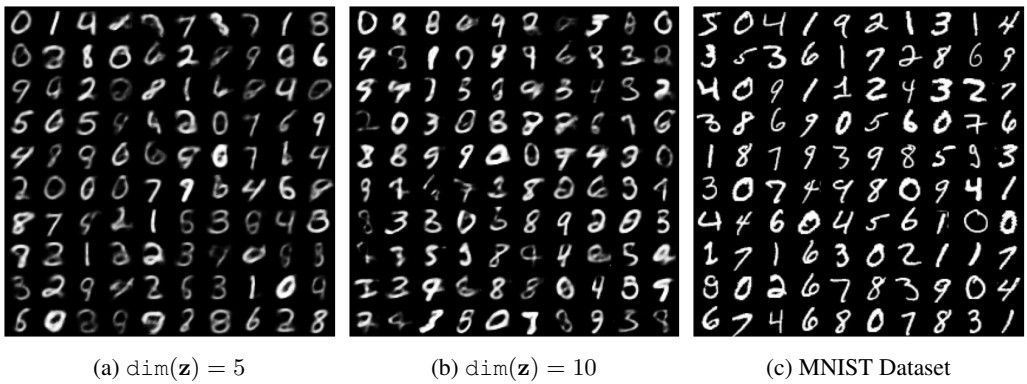

(a) $\texttt{dim}(\mathbf{z}) = 5$

(b) $\texttt{dim}(\mathbf{z}) = 10$

(c) MNIST Dataset

Figure 11: Training data for the flow model. Figure (a) and (b) are synthetic MNIST samples from two implicit models with latent dimension 5 and 10. Figure (c) are samples from the original MNIST dataset.

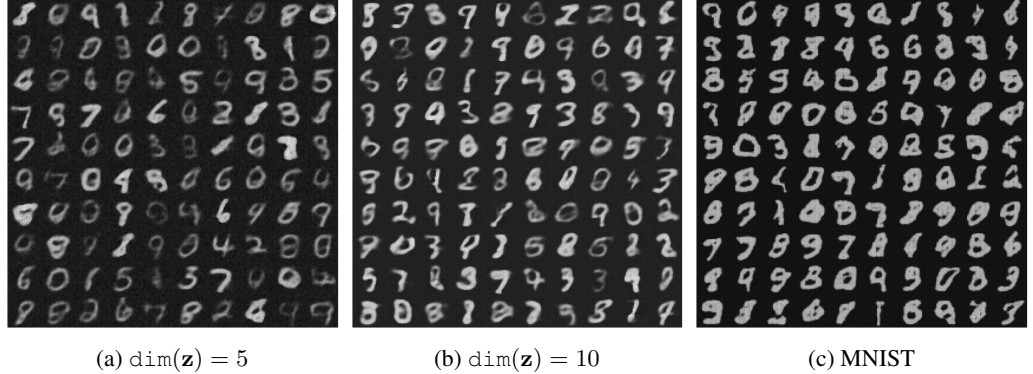

(a) dim($\mathbf{z}$) = 5          (b) dim($\mathbf{z}$) = 10          (c) MNIST

Figure 12: Samples from our methods. Figure (a) and (b) are samples flow models trained on synthetic MNIST data with intrinsic dimension 5 and 10. Figure (c) are samples from a flow model that trained on original MNIST data.

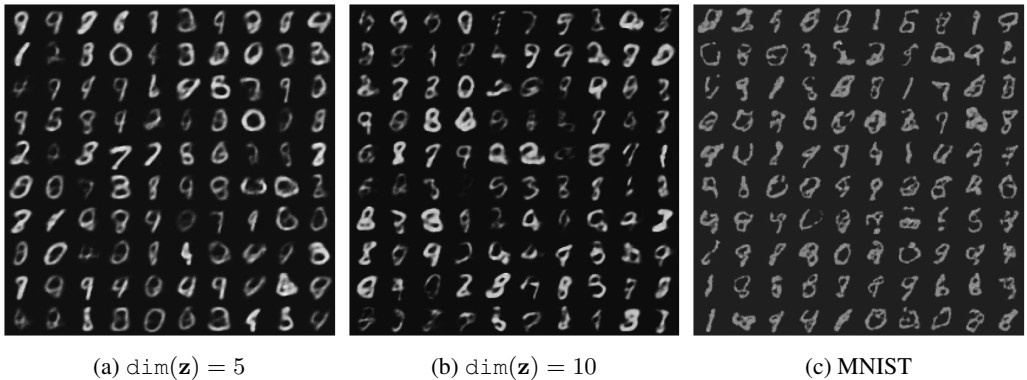

(a) dim($\mathbf{z}$) = 5          (b) dim($\mathbf{z}$) = 10          (c) MNIST

Figure 13: Samples from traditional non-volume preserving flow models with fixed Gaussian prior.

