# OpenReview forum: "On the Latent Space of Flow-based Models"
_ICLR.cc/2021/Conference — Reject_

### Official Review · AnonReviewer2 · 2020-10-27

**Rating:** 5
**Confidence:** 5

**Review:**

Pros:
1. this work propose to learn a manifold prior, by doing so, it can be used to improve the generation and representation quality.
Intrinsic dimension is applied in the method.
2. to fix the ill-defined KL, the authors proposed to use a "bridge" distribution, so that Q and P can have overlap on their support.


Cons:
even though I love the idea of this work, in the experiment section, the authors fail to compare their method with other flow-based method with quantitative results.
Since the authors claim that their method can improve the generation and representation quality, without any comparison, it lacks of evidence.

---

> ### Author Response · Authors · 2020-11-17
> **Envidence provided**
>
> We thank the reviewer for their positive feedback and address their comment by summarizing our extensive (and now updated) set of experimental evidence:
>
>    1. We demonstrate that our method enables flow models to learn the distribution that supports a Manifold whereas traditional flow models are unable to. In the revised paper, we provided extensive comparisons for
>     * 2d toy problems: Figure 1 and Figure 8 (our models have better sample/representation qualities);
>     * 3d manifold problem: Figure 4 and Figure 10 (our models have better sample/representation qualities);
>     * 1024d fading square dataset: Figure 5 (traditional flow fails to learn the data distribution);
>     * Synthetic and real MNIST data: Figure 12 and 13 (our models have better sample quality);
>
> 2.  Learn the intrinsic dimension of the target data distribution. We are not aware of any generative models that have this ability.

---

### Official Review · AnonReviewer4 · 2020-10-27
**A clean approach but lack of empirical evidence.**

**Rating:** 6
**Confidence:** 4

**Review:**


### Summary:

This paper uses the idea of spread KL divergence to learn densities for data that lies on a lower-dimensional manifold. The proposed method does not require the knowledge of this dimension and even offers a mechanism to estimate this intrinsic dimensionality. The authors validate their ideas on several toy-experiments.

### Strength:

The paper's idea is elegant; it is extremely well written for most parts, and the visualizations are informative. Authors use several compelling examples and make their case favorably.  The notation is pinpoint, and the paper is overall an easy read.

### Concerns:

One primary problem with the paper is the reinvention of the connection between MLE and KL divergence minimization in Z-space. This is well-documented (see [Papamakarios et al. 2017, Appendix A]; [Papamakorios et al. 2019, Appendix A])--not only authors skipped these citations, but the paper in its current form depicts this as a novel contribution (see Section 1, paragraph 2.)

I think the extension of this method to large-scale problems is not as trivial. The use parameters of the lower-triangular A scale as $O(D^2)$ can quickly become prohibitive.  Similarly, posthoc computation of eigenvalues of $AA^T$ can become expensive. Further, as noted by the authors in their experiments with MNIST, the real-world data may lie on a noisy lower-dimensional manifold--a modeling challenge that the current theory does not account for. I believe the easiest way to convince me would be to do experiments similar to Brehmer and Cranmer, 2020 on Style-GAN images (see Brehmer and Cranmer, 2020, Section 4, subsection E.) Alternatively, authors can explain why scalability is not an issue and provide other evidence for real-data performance.

#### Minor concern:

In Cunnigham et al. 2020, the authors consider learning densities for data that lie on lower-dimensional manifolds using normalizing flows. Their approach starts with a low-dimensional latent variable and uses affine transformations to change dimensions; they also use additive Gaussian noise such that the final density is in the ambient space. I believe there are interesting parallels in this work and their work--the authors should consider drawing out this comparison in the revision.

### References:

- Edmond Cunningham, Renos Zabounidis, Abhinav Agrawal, Ina Fiterau, Daniel Sheldon. Normalizing Flows Across Dimensions. ICML workshop, 2020.
- George Papamakarios, Theo Pavlakou, and Iain Murray. Masked autoregressive flow for density estimation. NeurIPS, 2017.
- George Papamakarios, Eric T. Nalisnick, Danilo Jimenez Rezende, Shakir Mohamed, and Balaji Lakshminarayanan. Normalizing flows for probabilistic modeling and inference. http://arxiv.org/abs/1912.02762. 2019

### Updates after the rebuttal

I found the approach clean and believe it has some merit among competing approaches. However, as detailed in discussions, I am skeptical of the scalability of the approach. I understand that an approach with scale limitations is acceptable and believe authors can benefit from an honest discussion about tradeoffs--which is currently missing. Further, I would suggest adding non-trivial experiments in the same vein as the ones in $\mathcal{M}-$flow paper. I think with these revisions, the paper can make for a welcome addition to the flows-for-density-on-manifold-literature.

---

> ### Author Response · Authors · 2020-11-17
> **Scalability is not a large concern, clarified the contribution in the revised paper.**
>
> We gratefully thank the reviewer for the valuable reviews, here are our attempts to address the concerns.
>
> 1.  We thank the reviewer again for the references, which we now cite in our updated manuscript, and refer also to our reply to Reviewer 3 (point 1) on this point.
> 2.  To determine the rank of a degenerative Gaussian, we need only D parameters in principle. We parameterize the triangular matrix of the covariance in order to learn the rotation of the prior as well. Alternatively, one can design a prior with D diagonal parameters such that the last layer of the flow function is a rotation matrix, resulting in equivalence between the two considered models. This observation highlights the fact that our method simply `moves' a number of parameters from the flow function to the prior. We further note that traditional flow models typically require large parameter counts, compared to the parameter count for the flow function, $\frac{D(D-1)}{2}$ this can be regarded as negligible. In summary, we believe the issue of scalability is not a large concern.
> 3.  We concede that our approach only works for data distributions that lie on a manifold. For distributions that alternatively lie around a manifold, we would propose the use of alternative modelling routes such as VAE. We leave related experimentation on large scale data to future work.
>
> 4.  To determine the intrinsic dimension of the target distribution, we do require a SVD of the covariance matrix. However, this is required only exactly **once** post-training and is not required during the training process. In practice we note that our strategy requires only a few seconds, for even the largest problems considered (in terms of dimensionality). This is negligible with respect to flow training time in all cases.
>
> 5.  We thank the reviewer for highlighting the related workshop paper. We consult the paper carefully and note that their work attempts to model distributions that lie **around** a manifold rather than **on** a manifold (c.f. the last sentence of their section 2). Therefore, their model is absolutely continuous (i.e. allows for density function and maximum likelihood learning) and is more closely aligned with a latent variable model e.g. VAE. A further minor difference is that their approach requires to pre-specify the latent dimension size, in contrast to our work. We add relevant discussion in our revised paper (section 8) and thank the reviewer for their drawing this work to our attention.

---

### Official Review · AnonReviewer3 · 2020-10-29

**Rating:** 4
**Confidence:** 4

**Review:**

This paper proposes a modification to the latent distribution of a flow model, replacing the commonly used full-rank Normal with a low-rank one which has the form N(0, AA^T). To train this degenerate model, the spread divergence from Zhang et al. (2020) is used and some approximation is made (e.g., ignoring the first entropy term). Experiments on toy data (with 1d intrinsic dimension) and mnist digits generated from a GAN (with 5-10 intrinsic dimension) demonstrated that the proposed model can identify the intrinsic dimension through the rank of AA^T.

##### Originality & Significance
As far as I know, the proposal of modeling the latent distribution of a flow model with a low-rank Normal is original .
Other parts of the paper are more like combination of existing methods and known results (e.g., the invariance of KL under invertible transformation, spread divergence).

##### Clarity
The clarity is generally good, although I don't feel the result in equation (4) is worth highlighting in the introduction. It can be a bit misleading as this is a well-known result.

##### Strengths
* Figure 1(c) is very illustrative as it shows how the distribution is curling up in the latent space to behave more like that it has two intrinsic dimensions, and the conclusion that neural network capacity is wasted by doing this.
* The experiment results are good and it is clever to use the data generated from a GAN to control the intrinsic dimension of image data, although the data is still very toy.

##### Weaknesses
* The authors mentioned several times that they "establish a connection between MLE and KL minimization in Z space". I guess they are not aware of this result thus trying to make it a contribution of this work. However, the invariance of KL under reparameterization is a well-known result, and it immediately follows that doing MLE = minimizing KL in X space = minimizing KL in Z space.
* The entropy term in equation (12) can only be ignored given two conditions
 * g is volume preserving, so that the log jacobian term disappears.
 * The variance of K is close to zero.
did the authors all use volume preserving flows in experiments? And the second condition can never be true in practice. In the paper an experiment that approximates this term has been conducted to justify the choice. But the approximation is very rough and I'd encourage the authors to try more accurate gradient estimators for the entropy term. For example, there is nonparametric score estimators (https://github.com/miskcoo/kscore) which can be easily plugged in here if you want to make a fast attempt, or you could try parametric estimators like sliced score matching (which requires training a neural network).
* All experiments are sort of toy and has only <10 intrinsic dimensions. Among them the most complex data is MNIST. Therefore, it is  a question if there exist other easier methods that could also find out the intrinsic dimension given the problem is very simple (e.g., nonlinear PCA).

---

> ### Author Response · Authors · 2020-11-17
> **Clarified the contribution, gradient estimators and nonlinear PCA**
>
> Thanks to the reviewer for the valuable suggestion, we hope to resolve the concerns with the following arguments.
> 1.  We do not intend to claim that a connection between KL in X space and KL in Z space is our novel contribution and apologize for any confusion that arose. We update our manuscript to make this known result clear and include citations to relevant work, as suggested.
>
> 2. Yes, we used the volume-preserving flow and it is not a limitation for the following reasons, see the reply to Reviewer 5 (point 1).
>
> 3.  We thank the reviewer for their suggestion regarding gradient estimators for the entropy term. Our currently employed approximation (see Append B.3) attempts to provide evidence highlighting the intuition that the entropy term does not affect the training objective, such that it may be safely ignored. Our experiments provide evidence to further support the argument. We note that kernel based estimators, suggested by the reviewer, are either computational expensive or have performance directly dependent on data dimension. Further, the dimension of the latent space $Dim(Z)=Dim(X)$ is typically very large in cases that we consider. In summary, we concede that better estimators may exist for the entropy term, however we consider further compute heavy exploration of this idea somewhat out of scope and leave it to future work. As a result of the reviewer feedback, the revised paper (see Appendix B.3), now reports that removal of the entropy term is an upper bound for the true objective. We believe this presents an interesting future direction; to assess to level to which the distance between the bound and the true objective depends on the related flow function when spread noise is small.
>
> 4.  The focus of our work is enabling flow models to fit the distributions that lie on manifolds and identify the intrinsic dimension as a byproduct. We are not aware of any generative models that can identify the intrinsic dimension of the data manifold. We offer the idea that our model may be viewed as an non-linear PCA where the non-linear component is solved by an invertible function. We further note that the probabilistic (generative) version of nonlinear PCA is just a VAE. Where a VAE is an absolutely continuous model and yet is not able to learn data distributions that lies on a manifold (the distribution is singular and does not have a density function). Another popular non-linear PCA is kernel PCA, where there lacks a principled route to select (or learn) kernels for complex data distribution. We hope these arguments could resolve the reviewer's concerns.

---

### Official Review · AnonReviewer1 · 2020-10-29
**Good work on training flow models on low-dimensional manifolds but lack some theoretical guarantees**

**Rating:** 5
**Confidence:** 4

**Review:**

This paper proposes a new method to train flow models on data from low dimensional manifolds embedded in high dimensional ambient spaces. The basic idea is based on minimizing the KL divergence in the latent space, which is equivalent to maximizing expected log-likelihood over the data distribution. Since the KL between two low-dimensional distributions are often undefined, authors propose to use spread divergence as a surrogate. Experiments demonstrate that the authors' technique can successfully model the distributions on low-dimensional manifolds without knowing the manifold beforehand. In particular, the proposed method can recover the intrinsic dimensionality of data manifolds.

#### Pros

* The proposed method simultaneously learns the manifold of a data distribution and the density on it, without the need of manually specifying the manifold.

* The method can give the intrinsic dimensionality of the data manifold. Authors have carried out extensive experimental study to verify this claim using both toy and natural datasets. The method can also be used as an approach for dimensionality reduction.

#### Cons

* The theoretical part is weak. As admitted by the authors, their analysis is not applicable to data distributions that are not absolutely continuous w.r.t. the Lebesgue measure in the ambient space. A stronger analysis should prove that $H(Z_\mathbb{Q} + K)$ is a constant for fixed $\sigma_Z$ and volume-preserving $g$.

* Experiments only show that the flow model can learn the manifold correctly. However, data points on the manifold can distribute differently. It seems all experiments in this paper assume that data points lie uniformly on the manifold. It is necessary to see experimental results where data have different densities in different regions of the manifold to check whether the proposed flow model can fit them accurately.

* The proposed method requires using a volume-preserving flow model. This is a big limitation to the expressivity of model architectures.

* In Figure 1, $p_d$ and $p_X$ should be swapped in the legend.

--------------
Post-rebuttal

I would like to thank the authors for the response. However, my concern on the theoretical part remains. For densities within the manifold, I suggest reviewers consider experiments similar to those in [1][2]. I also agree with R3, R4 on the reinvention of the connection between KL in the data space and KL in the latent space. In addition, I agree with R1, R3, R4 on the scale of their experiments and scalability of the approach. As such, I will lower my score from 6 to 5.


[1] Mathieu, Emile, and Maximilian Nickel. 2020. “Riemannian Continuous Normalizing Flows.” arXiv [stat.ML]. arXiv. http://arxiv.org/abs/2006.10605.
[2] Lou, Aaron, Derek Lim, Isay Katsman, Leo Huang, Qingxuan Jiang, Ser-Nam Lim, and Christopher De Sa. 2020. “Neural Manifold Ordinary Differential Equations.” arXiv [stat.ML]. arXiv. http://arxiv.org/abs/2006.10254.

---

> ### Author Response · Authors · 2020-11-17
> **Additional content provided to address the concerns.**
>
> We thank the reviewer for value suggestions, we provided some additional content in our revised paper to address the concerns.
>
> 1.  "A stronger analysis should prove that the entropy is a constant... ."  On the contrary, we note that $\textrm{H}(Z_{Q}+K)$ is not a constant for fixed and volume-preserving $g$. We provide a counterexample in our revised paper Appendix B.1. Additionally, we show that leaving out the entropy term during training corresponds to minimizing an upper bound of the true objective (see Appendix B.3).
>
> 2.  We appreciate the idea to experiment with varying manifold region densities and thank the reviewer for the insightful suggestion. Accordingly we update our manuscript to highlight the fact that our method can accurately capture density allocation for toy problems (c.f. updated Fig. 9). For high-dimensional problems, e.g. image data, we concede that density allocation is challenging to accurately diagnose and note this to be a valid future generative modelling research direction.
>
> 3. We think volume-preserving is not a limitation in our case. Please consult our reply to R5 (point 1) for the concern relating to volume-preserving flows.
>
> 4.  We thank the reviewer for highlighting minor errors and update our manuscript accordingly.

---

### Official Review · AnonReviewer5 · 2020-11-06
**Some interesting ideas, but lacks sound justification.**

**Rating:** 5
**Confidence:** 3

**Review:**

This paper proposes a new method of training flow models, instead of minimizing KL divergence in the data space X, the paper proposes to minimize the KL divergence in the latent space Z. However, the problem of this is dimension mismatch, so the KL divergence is ill-defined. The proposed solution is to add noise, such that the KL divergence can be well defined again. In addition, this paper proposes learning a low dimensional prior p(Z) by a Gaussian distribution with low rank covariance matrix.

Pro

The proposed method addresses a very important concern about flow models, which is that the latent space dimension has to be identical to the data space dimension.

The paper proposes learning the dimensionality of the latent space (prior) distribution p(Z) by parameterizing it as a Gaussian with low rank covariance. This seems like a reasonable design choice.

The experiments show that for simple synthetic data, the proposed approach is able to recover the true dimensionality of the data.

Con:

The major caveat is that the method seems to only work for volume preserving flow models (since the term 1 of Eq (12) can only be approximately neglected if the flow is volume preserving); most modern flow architectures are not volume preserving. I think this is quite a major limitation that needs to be explicitly stated when introducing the scope of the work.

It is unclear why minimizing the KL divergence in latent space Z is better than minimizing the KL divergence in data space X (which is current standard practice)

I think the severity of the dimension mismatch problem is somewhat overstated. If one applies standard training procedure for flow models (i.e. maximum likelihood on data space X) instead of the alternative proposed in the paper (minimizing KL divergence in latent space Z), then the problem in Figure 1 will not occur? I think this is because the dimension of the model distribution is greater than the true data distribution (which is conjectured to lie on a manifold), so log likelihood is well defined.

The experiments are somewhat weak. Only synthetic datasets are used. If the claimed benefit is better sample generation quality, then at least readers would expect standard benchmarks such as CIFAR.

Minor issues:

The title is not very informative

The major contribution of the paper is not stated early on (for example, the introduction seems like a background section).

---

> ### Author Response · Authors · 2020-11-17
> **Volume preserving is not a limitation and KL in Z space is equivalent to KL in X space.**
>
> We thank the reviewer for their positive feedback,  we believe that we could  address the concerns with the following arguments.
>
> 1.  We note that non-volume preserving flows usually contain fixed priors whereas our method concerns volume-preserving flows with a learnable prior (i.e. the prior volume can change). Both approaches have an ability to learn the volume of the target distribution and thus have the same expressive power. See the last paragraph of section 6, term 1 for further details. Popular modern flow architectures, such as affine couplings, can further easily be normalized to become volume-preserving, thus extending the scope and applicability of our proposed approach. See Appendix C for an example. In summary,  we do not consider the volume-preserving nature of our approach to result in large limitations.
>
> 2.  In section 1, we show that minimizing KL divergence in Z space is **equivalent** to minimizing KL divergence in X space. See also Appendix A and equation 24 for a detailed derivation. Therefore, we believe the reviewer's comments regarding dimension mismatch to be misplaced. Furthermore, when the model intrinsic dimension is strictly larger than the data distribution intrinsic dimension, the data will have measure zero under the model, leaving KL divergence (and MLE) ill-defined, so the issues still exists. Dimension mismatch problem is also the major motivation of [Wasserstein GAN ](https://arxiv.org/pdf/1701.07875.pdf). Therefore, we think the severity of the dimension mismatch problem is not overstated.
>
> 3. Our claims are that our method can: 1. model the data distribution that lies on one manifold and 2. identify the intrinsic dimension of the data. Our extensive experiments on multiple data distributions provide comprehensive evidence in support of these claims. We additionally show promising results on image distributions (MNIST) where challenging real-world properties result in our base assumptions being unsatisfied.  In our revised paper, we provide the comparisons of sample quality with a traditional flow model, see Figure 12 and 13 for details, this gives us some preliminary evidence that our models can achieve better sample quality on MNIST data. We hope that our contributions can prove useful and inspire other researchers to extend our work to further accommodate other challenging real-world data.
>
> 4.  We thank the reviewer for providing writing-style feedback and will endeavor to make our contributions clear at an earlier stage of the manuscript.

---

### Author Response · Authors · 2020-11-17
**Summary of the updates in the revised paper**

We added the following content in the revised paper:
1. Section 1: We clarified the connection between KL in X space and Z space is not our novel contribution.
2. Figure 5: We showed that traditional flow model fails to learn the fading square dataset.
3. Section 8: We added some reference.
4. Appendix B.1: We added an example to show that the entropy $H(g(X_d)+K)$ depends on the volume-preserving function $g$.
5. Appendix B.3: We showed that leaving out the entropy term corresponds to minimizing an upper bound of the true objective.
6. Appendix C.2: We added experiments to show that our model can capture the density allocation on the manifold for toy problems.
7. Appendix C.5.2: We added comparisons to the samples from traditional flow models.

---

### Decision · Program_Chairs · 2021-01-07
**Final Decision**

**Decision:**

Reject

**Comment:**

The paper proposes an algorithm for training flow models by minimizing the KL divergence in the latent space Z. The paper addresses an important problem in training flow models. However, some major concerns remain after the discussion among the reviewers. The scale of the experiments and the scalability of the approach appear limited in the current version of the paper. Moreover, the applicability of the current theoretical analysis to general distributions is quite limited.